## RESEARCH ARTICLE

# Association between uric acid and non-alcoholic fatty liver disease in patients with hypertension

**Yan Zhao** [1], **Zhenwei Wang**[2], **Yang Liu**[1], **Ting Yang**[1], **Zhuang Li**[1], **Ao Gao**[1], **Lei Cao**[1], **Chongyang Ma**[3]*

**1** Preventive Medicine Department, Dongfang Hospital, Beijing University of Chinese Medicine, Beijing, China, **2** Department of Cardiology, The First Affiliated Hospital of Zhengzhou University, Zhengzhou, China, **3** School of Traditional Chinese Medicine, Capital Medical University, Beijing, China

* machongyang@live.com

## Abstract

Serum uric acid (UA) is linked to non-alcoholic fatty liver disease (NAFLD), but its role in hypertensive populations remains unclear. This cross-sectional study investigated their association in 1,058 patients with hypertension. Multivariate logistic regression analysis confirmed that UA was independently correlated with NAFLD, whether as a continuous variable or a categorical variable. According to the fully adjusted model, the risk of NAFLD increased by 0.2%, 347.2% and 91.7% for each unit increase in UA, $\text{Log}_{10}$UA and LnUA, respectively (P < 0.05). Multivariate stratified analysis revealed that UA increased the risk of NAFLD in specific subgroups, including males, individuals aged 70–79 years, non-smokers, those without diabetes, and obese individuals (P < 0.05). Receiver operating characteristic (ROC) analysis indicated that UA could not only predict the occurrence of NAFLD but also improve the predictive value of the baseline model for NAFLD (UA, AUC: 0.588; baseline model, AUC: 0.770; baseline model + UA, AUC: 0.772). In conclusion, UA is significantly associated with NAFLD in patients with hypertension and may serve as a predictive risk indicator.

## 1. Introduction

Non-alcoholic fatty liver disease (NAFLD) is a highly prevalent metabolic disorder that is currently recognized as the most common chronic liver disease worldwide and poses a significant global health challenge [1]. This condition follows a progressive course, potentially advancing from hepatic steatosis to fibrosis, cirrhosis, and ultimately hepatocellular carcinoma. In addition to its impact on the liver, NAFLD exerts substantial systemic effects, contributing to the development and progression of various extrahepatic complications [2]. Accumulating evidence indicates that obesity, dyslipidemia, and diabetes are key risk factors for NAFLD, particularly among individuals with nocturnal hypertension [3]. Studies have consistently shown a significantly greater prevalence of NAFLD in hypertensive individuals than in normotensive

**Data availability statement:** The data that support the findings of this study are publicly available from the Dryad repository. These data can be accessed directly, without any special permissions, via the official repository: https://doi.org/10.5061/dryad.7d7wm3809.

**Funding:** This research was sponsored by the National Key Research and Development Program of China (No. 2024YFC3506301), and Wang Qi's National Master of Traditional Chinese Medicine Inheritance Studio (DFRCYJ-2024A-001). The funders had no role in study design, data collection and analysis, decision to publish, or preparation of the manuscript. This statement has been updated in the manuscript file. The authors received no specific funding for this work. And no author received a salary from these funders for conducting this study.

**Competing interests:** The authors have declared that no competing interests exist.

individuals [4,5]. This strong association underscores the urgent need to identify risk factors for NAFLD among patients with hypertension. Understanding modifiable determinants of NAFLD in this population can inform targeted preventive interventions and enhance integrated, multidisciplinary management strategies.

NAFLD is characterized by lipotoxic liver injury triggered by metabolic dysregulation [6]. Current research indicates that serum UA, as a metabolic biomarker, plays multiple roles in the pathogenesis of NAFLD, involving key pathways such as metabolic dysfunction, oxidative stress, the inflammatory response, and insulin resistance [7]. Clinical studies also suggest that UA may serve as a clinically meaningful predictor of NAFLD. Zhou *et al.* conducted a large cohort study involving 2,049 participants and revealed that the incidence of NAFLD increased across quartile groups (Q1 to Q4) with increasing UA levels, at 5.27%, 10.88%, 15.03%, and 19.18%, respectively [8]. Furthermore, a systematic review and meta-analysis by Sun *et al.* synthesizing 36 cross-sectional studies, 13 cohort studies, and 1 case–control study revealed a positive correlation between UA and NAFLD. Compared with individuals with lower UA levels, those with higher UA levels had approximately 1.88 times greater risk of NAFLD [9]. Another study in diabetic populations confirmed a significant association between uric acid and NAFLD. The likelihood of NAFLD progressively increased from the second quartile to the fourth quartile of UA levels. Even after adjusting for confounding factors such as age, sex, body mass index (BMI), and other metabolic components, the probability of NAFLD remained significantly elevated in the fourth quartile [10].

However, a comprehensive review of the literature revealed that the clinical utility of UA for NAFLD risk assessment in hypertensive populations remains understudied. Therefore, this study aimed to determine whether elevated UA levels are significantly associated with increased NAFLD risk in hypertensive patients and to examine the consistency of this association across different demographic subgroups. Our findings may provide new insights and an evidence-based framework for the risk stratification of NAFLD in patients with hypertension.

## 2. Methods

### 2.1 Study population

In this single-center, cross-sectional, retrospective study, 1592 individuals aged 40–79 years who received health examinations were enrolled from Wuhan Union Hospital. The inclusion criteria for patients were as follows: (1) clear diagnosis of hypertension [11] and (2) availability of diagnostic information for NAFLD. The exclusion criteria were as follows: (1) individuals who reported a history of known liver disease, including viral, autoimmune and drug-induced liver disease; (2) individuals with a diagnosis of acute illness, renal insufficiency (estimated glomerular filtration rate < 60 mL/min/1.73m$^2$), or active cancer (defined as self-reported history of cancer diagnosed or treated in the past 6 months); (3) individuals with oral or injectable steroids and those with missing biochemical measurements or medical history interview records; and (4) excessive alcohol consumption (defined as > 210 grams/week for men and > 140 grams/week for women). Ultimately, 1058 individuals were included in this study. This study utilized anonymized data obtained from publicly accessible

Dryad databases (https://doi.org/10.5061/dryad.7d7wm3809) [12]. We accessed and downloaded the dataset for research purposes in February 21, 2025. All data were de-identified prior to public release. The authors had no access to information that could identify individual participants during or after the data collection. The original research protocol was approved by the Institutional Review Board of Tongji Medical College, Huazhong University of Science and Technology (S155). As this retrospective analysis involved exclusively deidentified, preexisting data in the public domain, additional ethical approval was not required per institutional guidelines and prevailing ethical standards. Furthermore, the study design complied with all relevant provisions of the Declaration of Helsinki. Given the retrospective nature of the investigation and complete anonymization of all patient data, the ethics committee granted a waiver of informed consent. All the data were handled in strict accordance with institutional data protection policies and privacy regulations.

## 2.2 Data collection and definition

The data included in this study primarily included demographic information, anthropometric measurements, medical history, and serum markers. Each participant completed a questionnaire to gather self-reported data regarding sex, age, tobacco use, alcohol use and medical and medication history. Age was categorized into four groups: 40–49 years, 50–59 years, 60–69 years and 70–79 years. Tobacco use was defined as either a current smoker or former smoker, while alcohol use was defined as a current drinker or former drinker. Diabetes was defined on the basis of participants' self-reported history of diabetes or their use of hypoglycemic medications. Similarly, hypertension was identified through a self-reported history of hypertension or the use of oral antihypertensive drugs. BMI was calculated as weight in kilograms divided by height in meters squared (kg/m²), with a BMI of ≥ 24 classified as overweight or obese [13]. The serum markers measured included total cholesterol (TC), triglyceride (TG), high-density lipoprotein cholesterol (HDL-C), low-density lipoprotein cholesterol (LDL-C), alanine aminotransferase (ALT), aspartate aminotransferase (AST), UA and fasting blood glucose (FBG).

UA was the exposure factor in this study. Owing to the nonnormal distribution of UA levels, base-10 logarithmic ($Log_{10}$) and natural logarithmic (Ln) transformations were applied to normalize the data. Additionally, based on the quartiles of UA, the data were divided into Q1 (n = 265), Q2 (n = 265), Q3 (n = 264) and Q4 (n = 264), with Q1 ≤ 313.2 µmol/L, 313.2 µmol/L < Q2 ≤ 372.1 µmol/L, 372.1 µmol/L < Q3 ≤ 434.5 µmol/L, and Q4 > 434.5 µmol/L.

## 2.3 Assessment of NAFLD

The participants underwent conventional abdominal ultrasound examinations performed by trained technicians via a Philips IU22 system (Philips Healthcare, Inc.). NAFLD diagnosis was based on the distinctive ultrasonic features of diffuse hepatic steatosis, characterized by (1) anterior beam enhancement with increased echogenicity, (2) progressive posterior beam attenuation, and (3) reduced clarity of hepatic structures. Additionally, cases of hepatic steatosis attributable to excessive alcohol consumption (daily intake >20 g for females, > 30 g for males) and other known causes of hepatic steatosis (e.g., hepatitis B virus [HBV] and hepatitis C virus [HCV] seropositivity, autoimmune and drug-induced liver disease) were excluded. The remaining cases of hepatic steatosis were classified as NAFLD, according to the guidelines for the diagnosis and management of NAFLD published by the Chinese Society of Hepatology [14].

## 2.4 Statistical analysis

In this study, all the statistical analyses were conducted via SPSS version 26.0. First, a normality test was performed on all continuous variables via the Shapiro–Wilk test. Continuous variables that conformed to a normal distribution are represented as the mean ± standard deviation, whereas those that did not conform to a normal distribution are represented as the median (quartile). For continuous variables following a normal distribution, one-way analysis of variance (ANOVA) was employed to assess differences among the four groups. For continuous variables that did not follow a normal distribution, the Kruskal–Wallis test was used to evaluate differences among the four groups. Categorical variables are expressed in terms of frequency (percentage), and differences between groups were tested via the chi-square test. The risk factors for

NAFLD were evaluated through univariate logistic regression analysis, and variables with a P value less than 0.05 were selected for multivariate logistic regression analysis. Three adjustment models were established: Model 1 adjusted for age only; Model 2 adjusted for age, sex, tobacco use, alcohol use, diabetes, BMI and overweight or obesity; and Model 3 adjusted for age, sex, tobacco use, alcohol use, diabetes, BMI, overweight or obesity, TG, TC, LDL-C, HDL-C, FBG, ALT and AST. Subgroup analyses were subsequently conducted to evaluate the multivariate stratified association between UA levels and NAFLD in patients with hypertension. Finally, receiver operating characteristic (ROC) curve analysis was performed to assess the predictive value of UA and the baseline model for NAFLD. All tests were two-sided, and a P value of less than 0.05 was considered statistically significant.

### 2.5 Model validation and robustness analyses

To ensure the reliability and robustness of the findings from the multivariable logistic regression model, we performed comprehensive validation and diagnostic procedures.

Internal Validation via Bootstrap Resampling: The stability of coefficient estimates was evaluated using bootstrap res-ampling (1,000 replicates). The 95% confidence intervals for reporting were directly generated from the percentile-based empirical distribution of the bootstrapped coefficients.

Model Diagnostics:

Multicollinearity: Variance Inflation Factors (VIFs) were calculated for all continuous independent variables by running a linear regression model. A VIF value ≥ 10 was considered indicative of severe multicollinearity.

Goodness-of-fit: The calibration of the final logistic model was assessed using the Hosmer-Lemeshow test.

Sensitivity Analysis for Outliers and Influential Points: To evaluate whether our conclusions were unduly influenced by extreme observations, we identified potential outliers and influential points based on the following criteria: absolute standardized residuals > 2.5, Cook's distance > 0.1, and leverage values > 2*(k + 1)/n (where k is the number of predic-tors, and n is the sample size). A sensitivity analysis was then conducted by refitting the primary multivariable model after excluding the identified observations.

## 3. Results

### 3.1 Clinical characteristics according to UA quartile groups

As shown in Table 1, the variables of sex, age, tobacco use, alcohol use, overweight or obesity, NAFLD, BMI, TG, TC, LDL-C, HDL-C and ALT levels were significantly different among the UA quartile groups (P < 0.05). Specifically, the Q4 group had a greater number of individuals who were drinkers and patients with NAFLD, as well as higher BMIs, TG, TC, LDL-C and ALT levels.

### 3.2 The correlation between UA and NAFLD

As shown in Table 2, the univariate logistic regression analysis indicated that sex, age, tobacco use, alcohol consumption, diabetes status, overweight or obesity status, BMI, FBG, TG, TC, LDL-C, HDL-C, ALT, AST, and UA were significantly associated with the risk of NAFLD (P < 0.05).

As shown in Table 3, the multivariate logistic regression analysis indicated that in Model 1, which was adjusted solely for age, UA was significantly associated with the risk of NAFLD, regardless of whether it was treated as a continuous or categorical variable (P < 0.05). In Model 2, which was adjusted for age, sex, tobacco use, alcohol consumption, dia-betes status, BMI, and overweight or obesity status, higher levels of UA continued to be significantly associated with an increased risk of NAFLD (P < 0.05). In Model 3, UA remained significantly associated with the risk of NAFLD, and for each one-unit increase in UA, Log10UA, and LnUA, the risk of NAFLD increased by 0.2%, 347.2%, and 91.7%, respectively

**Table 1. Clinical characteristics of study participants according to uric acid quartile groups.**

| Variable | Total | Q1 | Q2 | Q3 | Q4 | P value |
|---|---|---|---|---|---|---|
| N | 1058 | 265 | 265 | 264 | 264 | |
| Gender, n (%) | | | | | | <0.001 |
| Female | 238 (22.5) | 125 (47.2) | 66 (24.9) | 24 (9.1) | 23 (8.7) | |
| Male | 820 (77.5) | 140 (52.8) | 199 (75.1) | 240 (90.9) | 241 (91.3) | |
| Age, n (%) | | | | | | 0.006 |
| 40–49 years | 176 (16.6) | 34 (12.8) | 41 (15.5) | 49 (18.6) | 52 (19.7) | |
| 50–59 years | 443 (41.9) | 94 (35.5) | 118 (44.5) | 105 (39.8) | 126 (47.7) | |
| 60–69 years | 296 (28.0) | 93 (35.1) | 70 (26.4) | 78 (29.5) | 55 (20.8) | |
| 70–79 years | 143 (13.5) | 44 (16.6) | 36 (13.6) | 32 (12.1) | 31 (11.7) | |
| Tobacco use, n (%) | | | | | | <0.001 |
| Yes | 390 (36.9) | 65 (24.5) | 103 (38.9) | 114 (43.2) | 108 (40.9) | |
| No | 668 (63.1) | 200 (75.5) | 162 (61.1) | 150 (56.8) | 156 (59.1) | |
| Alcohol use, n (%) | | | | | | <0.001 |
| Yes | 374 (35.3) | 57 (21.5) | 96 (36.2) | 110 (41.7) | 111 (42.0) | |
| No | 684 (64.7) | 208 (78.5) | 169 (63.8) | 154 (58.3) | 153 (58.0) | |
| Diabetes, n (%) | | | | | | 0.652 |
| Yes | 418 (39.5) | 103 (38.9) | 111 (41.9) | 107 (40.5) | 97 (36.7) | |
| No | 640 (60.5) | 162 (61.1) | 154 (58.1) | 157 (59.5) | 167 (63.3) | |
| Overweight or obesity, n (%) | | | | | | <0.001 |
| Yes | 798 (75.4) | 174 (65.7) | 200 (75.5) | 209 (79.2) | 215 (81.4) | |
| No | 260 (24.6) | 91 (34.3) | 65 (24.5) | 55 (20.8) | 49 (18.6) | |
| NAFLD, n (%) | | | | | | <0.001 |
| Yes | 704 (66.5) | 149 (56.2) | 181 (68.3) | 172 (65.2) | 202 (76.5) | |
| No | 354 (33.5) | 116 (43.8) | 84 (31.7) | 92 (34.8) | 62 (23.5) | |
| BMI, kg/m$^2$ | 25.60 (24.00, 27.60) | 25.10 (23.50, 27.10) | 25.50 (24.00, 27.50) | 26.00 (24.10, 28.10) | 26.10 (24.50, 28.08) | <0.001 |
| SBP, mmHg | 136.00 (126.00, 146.00) | 138.00 (128.00, 148.00) | 136.00 (126.00, 146.00) | 136.00 (124.00, 145.75) | 134.00 (125.00, 142.00) | 0.087 |
| DBP, mmHg | 85.00 (78.00, 92.00) | 84.00 (77.50, 91.00) | 84.00 (76.00, 92.00) | 86.00 (78.00, 92.75) | 85.00 (78.00, 93.00) | 0.655 |
| FBG, mmol/L | 5.20 (4.76, 5.90) | 5.27 (4.79, 6.10) | 5.20 (4.73, 5.91) | 5.20 (4.70, 5.90) | 5.23 (4.80, 5.90) | 0.557 |
| TG, mmol/L | 1.52 (1.03, 2.29) | 1.22 (0.87,1.85) | 1.46 (1.04, 2.11) | 1.54 (1.03, 2.22) | 1.93 (1.28, 2.90) | <0.001 |
| TC, mmol/L | 4.30 (3.57, 5.09) | 4.37 (3.64, 5.13) | 4.06 (3.42, 4.85) | 4.16 (3.36, 4.90) | 4.51 (3.84, 5.26) | <0.001 |
| LDL-C, mmol/L | 2.51 (1.89, 3.15) | 2.57 (1.95, 3.20) | 2.30 (1.79, 3.03) | 2.48 (1.73, 3.07) | 2.72 (2.09, 3.33) | 0.001 |
| HDL-C, mmol/L | 1.05 (0.87, 1.24) | 1.15 (0.95,1.38) | 1.06 (0.88, 1.26) | 1.02 (0.87, 1.16) | 0.98 (0.83, 1.12) | <0.001 |
| ALT, U/L | 23.00 (16.00, 32.00) | 21.00 (14.00, 30.00) | 22.00 (16.00, 30.00) | 23.00 (17.00, 32.00) | 24.00 (17.00, 35.00) | 0.003 |
| AST, U/L | 21.00 (18.00, 27.00) | 21.00 (17.00, 26.00) | 21.00 (18.00, 26.75) | 22.00 (18.00, 28.00) | 21.00 (18.00, 27.00) | 0.688 |
| UA, μmol/L | 372.10 (313.20, 434.50) | 269.30 (241.30, 293.90) | 345.80 (330.35, 359.10) | 399.50 (386.53, 416.40) | 487.65 (456.00, 533.70) | <0.001 |
| Log$_{10}$UA | 2.56±0.11 | 2.42±0.06 | 2.54±0.02 | 2.60±0.02 | 2.70±0.05 | <0.001 |
| LnUA | 5.90±0.26 | 5.57±0.15 | 5.84±0.05 | 5.99±0.04 | 6.21±0.11 | <0.001 |

NAFLD, non-alcoholic fatty liver disease; BMI, body mass index; SBP, systolic blood pressure; DBP, diastolic blood pressure; FBG, fasting blood glucose; TG, triglycerides; TC, total cholesterol; LDL-C, low-density lipoprotein cholesterol; HDL-C, high-density lipoprotein cholesterol; ALT, alanine aminotransferase; AST, aspartate aminotransferase; UA, uric acid. Q1 ≤ 313.2 μmol/L, 313.2 μmol/L < Q2 ≤ 372.1 μmol/L, 372.1 μmol/L < Q3 ≤ 434.5 μmol/L, and Q4 > 434.5 μmol/L.

**Table 2. Binary Logistic regression analysis of NAFLD.**

| Variable | OR | 95% CI | P value |
|---|---|---|---|
| Gender | | | |
| Female | Ref | | |
| Male | 1.652 | 1.228-2.223 | 0.001 |
| Age | | | |
| 40–49 years | Ref | | |
| 50–59 years | 0.664 | 0.444-0.994 | 0.047 |
| 60–69 years | 0.485 | 0.318-0.738 | 0.001 |
| 70–79 years | 0.420 | 0.259-0.680 | <0.001 |
| Tobacco use | 1.493 | 1.138-1.959 | 0.004 |
| Alcohol use | 1.466 | 1.115-1.929 | 0.006 |
| Diabetes | 2.177 | 1.652-2.869 | <0.001 |
| Overweight or obesity | 3.883 | 2.898-5.203 | <0.001 |
| BMI | 1.364 | 1.283-1.449 | <0.001 |
| SBP | 1.000 | 0.992-1.008 | 0.996 |
| DBP | 1.006 | 0.995-1.018 | 0.282 |
| FBG | 1.354 | 1.210-1.515 | <0.001 |
| TG | 1.815 | 1.550-2.124 | <0.001 |
| TC | 1.240 | 1.099-1.399 | <0.001 |
| LDL-C | 1.162 | 1.007-1.340 | 0.040 |
| HDL-C | 0.272 | 0.173-0.426 | <0.001 |
| ALT | 1.044 | 1.031-1.056 | <0.001 |
| AST | 1.037 | 1.020-1.055 | <0.001 |
| UA | 1.004 | 1.002-1.005 | <0.001 |
| $Log_{10}UA$ | 18.655 | 5.739-60.642 | <0.001 |
| LnUA | 3.564 | 2.136-5.946 | <0.001 |
| UA as a categorical variable | | | |
| Q1 | Ref | | |
| Q2 | 1.678 | 1.177-2.392 | 0.004 |
| Q3 | 1.456 | 1.025-2.067 | 0.036 |
| Q4 | 2.536 | 1.745-3.687 | <0.001 |
| P for trend | | | <0.001 |

NAFLD, non-alcoholic fatty liver disease; BMI, body mass index; SBP, systolic blood pressure; DBP, diastolic blood pressure; FBG, fasting blood glucose; TG, triglycerides; TC, total cholesterol; LDL-C, low-density lipoprotein cholesterol; HDL-C, high-density lipoprotein cholesterol; ALT, alanine aminotransferase; AST, aspartate aminotransferase; UA, uric acid.

(OR = 1.002, 95% CI: 1.000–1.004, P = 0.027; OR = 4.472, 95% CI: 1.135–17.624, P = 0.032; OR = 1.917, 95% CI: 1.057–3.477, P = 0.032; respectively). Furthermore, the risk of NAFLD in the Q2 and Q4 groups was 1.513 and 1.643 times greater than that in the Q1 group, respectively (OR = 1.513, 95% CI: 1.006–2.274, P = 0.046; OR = 1.643, 95% CI: 1.043–2.589, P = 0.032; respectively).

## 3.3 Multivariate stratified analysis of the association between UA and NAFLD

As shown in Table 4, among males, the risk of NAFLD in the Q4 group was 1.782 times greater than that in the Q1 group. Additionally, for each one-unit increase in UA, the risk of NAFLD increased by 0.2% (P < 0.05). In the population aged

**Table 3. Correlation between UA and NAFLD.**

| | Model 1 | | | Model 2 | | | Model 3 | | |
|---|---|---|---|---|---|---|---|---|---|
| | OR | 95% CI | P value | OR | 95% CI | P value | OR | 95% CI | P value |
| Continuous variables | | | | | | | | | |
| UA | 1.003 | 1.002-1.005 | <0.001 | 1.003 | 1.001-1.004 | 0.001 | 1.002 | 1.000-1.004 | 0.027 |
| $Log_{10}UA$ | 14.639 | 4.435-48.323 | <0.001 | 8.480 | 2.310-31.121 | 0.001 | 4.472 | 1.135-17.624 | 0.032 |
| LnUA | 3.208 | 1.910-5.388 | <0.001 | 2.530 | 1.439-4.451 | 0.001 | 1.917 | 1.057-3.477 | 0.032 |
| Categorical variables | | | | | | | | | |
| Q1 | Ref | | | Ref | | | Ref | | |
| Q2 | 1.618 | 1.131-2.314 | 0.008 | 1.480 | 1.009-2.170 | 0.045 | 1.513 | 1.006-2.274 | 0.046 |
| Q3 | 1.387 | 0.974-1.976 | 0.070 | 1.102 | 0.752-1.615 | 0.619 | 1.072 | 0.699-1.643 | 0.751 |
| Q4 | 2.369 | 1.622-3.460 | <0.001 | 2.009 | 1.341-3.010 | 0.001 | 1.643 | 1.043-2.589 | 0.032 |
| P for trend | | | <0.001 | | | 0.003 | | | 0.049 |

Model 1: Adjusted for gender and age; Model 2: Adjusted for gender, age, tobacco use, alcohol use, diabetes, BMI, overweight or obesity; Model 3: Adjusted for gender, age, tobacco use, alcohol use, diabetes, BMI, overweight or obesity, TG, TC, LDL-C, HDL-C, FBG, ALT and AST.

UA, uric acid; NAFLD, non-alcoholic fatty liver disease; BMI, body mass index; TG, triglycerides; TC, total cholesterol; LDL-C, low-density lipoprotein cholesterol; HDL-C, high-density lipoprotein cholesterol; FBG, fasting blood glucose; ALT, alanine aminotransferase; AST, aspartate aminotransferase. Q1 ≤ 313.2 μmol/L, 313.2 μmol/L < Q2 ≤ 372.1 μmol/L, 372.1 μmol/L < Q3 ≤ 434.5 μmol/L, and Q4 > 434.5 μmol/L.

70–79 years, the risk of NAFLD in the Q3 and Q4 groups was 2.831 and 5.376 times greater than that in the Q1 group, respectively (P<0.05). Furthermore, for each one-unit increase in UA, $Log_{10}UA$, and LnUA, the risk of NAFLD significantly increased (P<0.05). Among non-smokers, the risk of NAFLD in the Q2 and Q4 groups was 1.662 and 1.860 times greater than that in the Q1 group, respectively (P<0.05). For each one-unit increase in UA, $Log_{10}UA$ and LnUA, the risk of NAFLD increased by 0.2%, 603.6% and 133.3%, respectively (P<0.05). Among participants with diabetes, the risk of NAFLD in the Q2 group was 2.109 times greater than that in the Q1 group (P<0.05). Among participants without diabetes, for each one-unit increase in UA, $Log_{10}UA$, and LnUA, the risk of NAFLD increased by 0.2%, 498.1%, and 117.4%, respectively (P<0.05). Among participants who were overweight or obese, the risk of NAFLD in the Q2 and Q4 groups was 1.734 and 0.643 times greater than that in the Q1 group, respectively (P<0.05). For each one-unit increase in UA, $Log_{10}UA$, and LnUA, the risk of NAFLD increased by 0.2%, 63.5%, and 155%, respectively (P<0.05).

### 3.4 The predictive value of UA and the baseline model for NAFLD

As illustrated in Fig 1A, the ROC analysis demonstrated that UA had a certain predictive value for the risk of NAFLD (AUC: 0.588, 95% CI: 0.552–0.624, P<0.001). Furthermore, Fig 1B also shows that both UA and the baseline model could predict the occurrence of NAFLD, with UA increasing the predictive ability of the baseline model (UA, AUC: 0.588; baseline model, AUC: 0.770; baseline model+UA, AUC: 0.772).

### 3.5 Robustness analysis and model diagnostics

As shown in Table 5, the stability of the key associations is supported by bootstrap analysis (1,000 replicates). The positive association between UA and NAFLD remained robust, with a narrow bootstrap 95% confidence interval (B=0.002, Bootstrap 95% CI: 0.000 to 0.004) that did not include zero. The associations of other significant variables, such as BMI and diabetes, were also stable in the bootstrap analysis.

The Hosmer-Lemeshow test indicated good calibration for the full model ($\chi^2$=11.668, P=0.167). Multicollinearity diagnostics confirmed no severe collinearity, as all VIFs were below 10, with the VIF for UA at 1.248. Based on predefined criteria, 55 observations were identified as potential outliers. In the sensitivity analysis performed after excluding these

**Table 4. Multivariate stratified analysis of the association between UA and NAFLD.**

| | Q2 vs Q1 | | Q3 vs Q1 | | Q4 vs Q1 | | UA | | Log₁₀UA | | LnUA | |
|---|---|---|---|---|---|---|---|---|---|---|---|---|
| | OR (95% CI) | P value | OR (95% CI) | P value | OR (95% CI) | P value | OR (95% CI) | P value | OR (95% CI) | P value | OR (95% CI) | P value |
| **Gender** | | | | | | | | | | | | |
| Male | 1.626 (0.983-2.689) | 0.058 | 1.107 (0.685-1.789) | 0.677 | 1.782 (1.069-2.971) | 0.027 | 1.002 (1.000-1.004) | 0.030 | 5.542 (0.920-33.392) | 0.062 | 2.104 (0.964-4.589) | 0.062 |
| Female | 1.591 (0.742-3.409) | 0.233 | 1.191 (0.369-3.847) | 0.770 | 0.808 (0.261-2.494) | 0.710 | 1.001 (0.998-1.005) | 0.471 | 5.409 (0.284-102.965) | 0.261 | 2.081 (0.579-7.483) | 0.261 |
| **Age** | | | | | | | | | | | | |
| 40–49 years | 0.602 (0.121-2.990) | 0.535 | 0.500 (0.100-2.504) | 0.399 | 0.221 (0.041-1.203) | 0.081 | 0.995 (0.989-1.001) | 0.106 | 0.008 (0.000-1.948) | 0.085 | 0.119 (0.011-1.336) | 0.085 |
| 50–59 years | 1.355 (0.692-2.653) | 0.376 | 0.881 (0.431-1.803) | 0.729 | 1.418 (0.699-2.877) | 0.334 | 1.002 (0.999-1.005) | 0.196 | 4.359 (0.414-45.927) | 0.220 | 1.895 (0.682-5.270) | 0.220 |
| 60–69 years | 2.238 (1.038-4.826) | 0.040 | 1.187 (0.537-2.623) | 0.671 | 2.138 (0.872-5.242) | 0.097 | 1.003 (1.000-1.007) | 0.060 | 13.098 (0.791-216.934) | 0.072 | 3.056 (0.903-10.343) | 0.072 |
| 70–79 years | 1.907 (0.735-4.947) | 0.185 | 2.831 (1.036-7.735) | 0.042 | 5.376 (1.777-16.266) | 0.003 | 1.007 (1.003-1.012) | 0.002 | 132.309 (3.540-4945.225) | 0.008 | 8.344 (1.732-40.213) | 0.008 |
| **Tobacco use** | | | | | | | | | | | | |
| Yes | 1.203 (0.537-2.696) | 0.654 | 0.849 (0.387-1.860) | 0.682 | 1.228 (0.512-2.944) | 0.646 | 1.001 (0.998-1.005) | 0.403 | 2.291 (0.127-41.378) | 0.575 | 1.433 (0.408-5.037) | 0.575 |
| No | 1.662 (1.019-2.711) | 0.042 | 1.159 (0.680-1.975) | 0.587 | 1.860 (1.065-3.250) | 0.029 | 1.002 (1.000-1.004) | 0.024 | 7.036 (1.339-36.963) | 0.021 | 2.333 (1.135-4.796) | 0.021 |
| **Diabetes** | | | | | | | | | | | | |
| Yes | 2.109 (1.052-4.226) | 0.035 | 1.274 (0.624-2.601) | 0.507 | 1.305 (0.600-2.840) | 0.502 | 1.002 (0.999-1.005) | 0.304 | 3.607 (0.279-46.637) | 0.326 | 1.746 (0.574-5.305) | 0.326 |
| No | 1.262 (0.749-2.126) | 0.383 | 0.969 (0.559-1.682) | 0.912 | 1.764 (0.986-3.157) | 0.056 | 1.002 (1.000-1.004) | 0.033 | 5.981 (1.094-32.685) | 0.039 | 2.174 (1.040-4.546) | 0.039 |
| **Overweight or obesity** | | | | | | | | | | | | |
| Yes | 1.734 (1.068-2.816) | 0.026 | 1.117 (0.700-1.781) | 0.643 | 1.869 (1.126-3.101) | 0.016 | 1.002 (1.000-1.004) | 0.044 | 8.635 (1.379-54.077) | 0.021 | 2.550 (1.1150-5.658) | 0.021 |
| No | 1.167 (0.550-2.476) | 0.688 | 0.743 (0.313-1.762) | 0.500 | 0.863 (0.353-2.108) | 0.746 | 1.001 (0.997-1.004) | 0.771 | 1.704 (0.096-30.182) | 0.716 | 1.260 (0.362-4.392) | 0.716 |

The subgroup analyses adjusted for gender, age, tobacco use, alcohol use, diabetes, BMI, overweight or obesity, FBG, TG, TC, LDL-C, HDL-C, ALT and AST in the model, except for the subgroup variables.

UA, uric acid; NAFLD, non-alcoholic fatty liver disease; BMI, body mass index; FBG, fasting blood glucose; TG, triglycerides; TC, total cholesterol; LDL-C, low-density lipoprotein cholesterol; HDL-C, high-density lipoprotein cholesterol; ALT, alanine aminotransferase; AST, aspartate aminotransferase. Q1 ≤313.2 μmol/L, 313.2 μmol/L <Q2 ≤372.1 μmol/L, 372.1 μmol/L <Q3 ≤434.5 μmol/L, and Q4 >434.5 μmol/L.

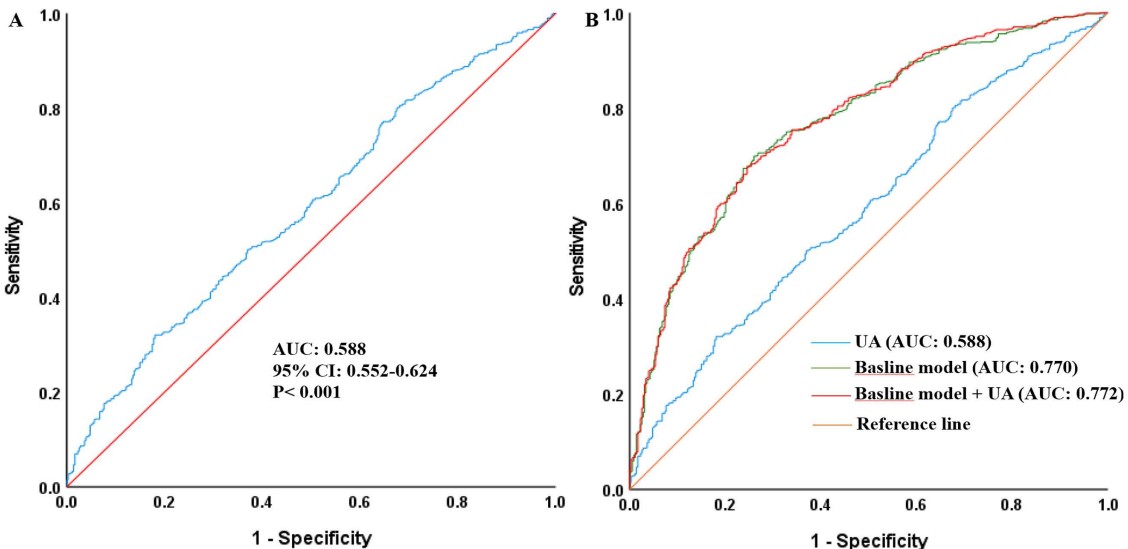

**Fig 1. ROC curves of UA and the baseline model for the prediction of NAFLD.** The baseline model included gender, age, tobacco use, alcohol use, diabetes, BMI, overweight or obesity, FBG, TG, TC, LDL-C, HDL-C, ALT and AST. AUC, Area Under Curve; ROC, receiver operating characteristic curve; UA, uric acid; NAFLD, non-alcoholic fatty liver disease; BMI, body mass index; FBG, fasting blood glucose; TG, triglycerides; TC, total cholesterol; LDL-C, low-density lipoprotein cholesterol; HDL-C, high-density lipoprotein cholesterol; ALT, alanine aminotransferase; AST, aspartate aminotransferase.

**Table 5. Bootstrap validation of the regression model.**

| Variable | Coefficient (B) | P | Bootstrap 95% CI |
|---|---|---|---|
| BMI | 0.188 | 0.001 | 0.103-0.299 |
| Overweight or obesity | −0.426 | 0.074 | −0.891-0.064 |
| ALT | 0.031 | 0.002 | 0.015-0.054 |
| AST | −0.012 | 0.245 | −0.043-0.010 |
| FBG | 0.096 | 0.166 | −0.044-0.251 |
| UA | 0.002 | 0.019 | 0.000-0.004 |
| TC | 0.083 | 0.635 | −0.761-0.698 |
| TG | 0.269 | 0.028 | 0.020-0.651 |
| HDL-C | −0.196 | 0.563 | −0.910-0.783 |
| LDL-C | 0.062 | 0.758 | −0.560-0.924 |
| Tobacco use | −0.168 | 0.382 | −0.552-0.210 |
| Alcohol use | 0.045 | 0.799 | −0.322-0.431 |
| Diabetes | −0.525 | 0.004 | −0.926-0.157 |

BMI, body mass index; ALT, alanine aminotransferase; AST, aspartate aminotransferase; FBG, fasting blood glucose; UA, uric acid; TC, total cholesterol; TG, triglycerides; HDL-C, high-density lipoprotein cholesterol; LDL-C, low-density lipoprotein cholesterol.

observations, the direction, magnitude, and statistical significance of the association between UA and NAFLD were unchanged (OR = 1.003, 95% CI: 1.001–1.005, P = 0.014), confirming that the primary finding is not driven by influential points.

## 4. Discussion

This cross-sectional study involving Chinese adult patients with hypertension revealed a significant association between UA and NAFLD risk. After comprehensive adjustment for all potential confounders, each unit increase in UA was significantly associated with an increased risk of NAFLD. To ensure the robustness of the identified association, we performed a series of rigorous model diagnostics. First, internal validation via bootstrap resampling (1,000 replicates) demonstrated that the association between UA and NAFLD remained precise and stable, with a narrow 95% confidence interval (0.000–0.004) that did not include zero. Second, sensitivity analysis confirmed that this association remained consistent after excluding potential outliers, indicating that the finding was not driven by influential observations. Finally, multicollinearity diagnostics secured the reliability of estimating independent effects, with all variance inflation factors below the threshold of concern. These results collectively support the high stability and reliability of the association between UA and NAFLD. Given the routine availability and low cost of UA testing in hypertension management, the minimal observed increase in AUC (from 0.770 to 0.772) following its addition to the model represents a cost-effective enhancement to NAFLD risk stratification. Therefore, serum UA can be considered a robust and practically valuable independent predictor of NAFLD in patients with hypertension.

As the end product of purine metabolism, UA is influenced by multiple factors, including dietary habits, demographic characteristics (such as age, sex, and race/ethnicity), and genetic predispositions. Studies indicate that elevated serum UA is an independent risk factor for cardiovascular and renal diseases, including hypertension, coronary artery disease, stroke, heart failure, and chronic kidney disease [15,16]. Since Leonardo et al. first identified UA as an independent predictor of NAFLD in 2002 [17], the association between UA and NAFLD has been extensively studied. Subsequent studies have confirmed this association, demonstrating that elevated serum UA levels correlate with an increased risk of NAFLD development and progression, thereby supporting its potential role as a predictive biomarker [18]. Wijarnpreecha et al. reported a significant increase in NAFLD risk among individuals with hyperuricemia in a meta-analysis encompassing 25 studies [19], further corroborating the positive correlation between hyperuricemia and NAFLD. Sun et al. conducted a systematic review and meta-analysis encompassing 50 studies and 2,079,710 participants [9], which was consistent with our finding of a positive association between elevated serum UA levels and NAFLD risk in patients with hypertension. However, the nature of this relationship is complex and context dependent. A Mendelian randomization analysis suggested a bidirectional association between serum UA and NAFLD, where NAFLD may increase UA levels, but a genetic predisposition to hyperuricemia does not increase NAFLD risk [20]. These inconsistencies may stem from methodological limitations, such as residual confounding in genetic studies and difficulties in detecting nonlinear relationships between exposure and outcome [21]. Our study revealed that the association between UA and NAFLD in hypertensive individuals was nonlinear. When the lowest quartile group (Q1) was used as a reference, NAFLD risk significantly increased with increasing UA levels starting in the Q2 group. However, no significant association was observed in the Q3 group, suggesting that the risk may plateau within this range. Finally, the strongest positive correlation was observed in the Q4 group, which presented the highest UA levels. This pattern indicates that the effect of UA on NAFLD is not a simple dose–response relationship, as even modest elevations in UA levels are harmful, while extremely high levels confer the greatest disease risk.

Furthermore, the association between UA and NAFLD appears to vary across different populations. For example, Bao et al. reported that UA is a significant predictor of NAFLD risk in non-obese postmenopausal women [22], while Duan and Chen et al. reported consistent positive correlations in both premenopausal and postmenopausal women in their respective studies [23,24]. Conversely, Fan et al. reported no significant association between UA and NAFLD risk in a female cohort of diabetic patients [25]. This result aligns with our subgroup analysis of hypertensive patients, where no significant relationship was observed between UA and NAFLD in hypertensive women, whereas a positive correlation was found in hypertensive men. These inconsistent findings may be attributed to heterogeneity in the study populations, variations in metabolic characteristics, or differences in the underlying pathophysiological mechanisms associated with NAFLD development.

Notably, as previous studies have not systematically evaluated the association between UA and NAFLD in hypertensive cohorts, our research provides novel insights into managing NAFLD on the basis of UA levels by specifically examining this relationship in hypertensive individuals. Our findings confirm that elevated uric acid (UA > 434.5 µmol/L) is an independent risk factor for NAFLD in hypertensive patients. Elevated UA levels may synergize with hypertension to accelerate hepatic steatosis and injury by promoting insulin resistance, oxidative stress, and inflammatory responses. For such patients, aggressive urate-lowering therapy offers dual cardiovascular protection while preventing or delaying NAFLD progression. Concurrently, these patients should be considered at high risk for NAFLD and undergo regular liver ultrasound screening alongside fibrosis assessment (e.g., FibroTest, liver stiffness measurement, magnetic resonance elastography). For women with hypertension, other factors (such as obesity and dyslipidemia) may be more significant drivers of NAFLD than uric acid is. These findings contribute to a deeper understanding of NAFLD pathogenesis in high-risk populations and provide clinical evidence for future research on strategies to prevent and manage NAFLD in hypertensive individuals.

Although the precise mechanisms by which UA contributes to NAFLD warrant further elucidation, increasing evidence indicates that UA promotes hepatic steatosis and injury through four interrelated pathological processes: oxidative stress, inflammatory activation, insulin resistance (IR), and dysregulated lipid metabolism. These mechanisms form a self-reinforcing cycle that drives NAFLD progression. At the cellular level, UA induces metabolic dysfunction via multiple pathways. Upon entering hepatocytes, UA promotes intracellular lipid accumulation by triggering oxidative stress and subsequent endoplasmic reticulum (ER) stress. This activates the unfolded protein response (UPR) pathway, modulates the expression of lipogenic proteins, and ultimately disrupts hepatic lipid homeostasis [26]. The pro-oxidant properties of UA further exacerbate metabolic dysfunction by increasing reactive oxygen species production and promoting lipid peroxidation. Concurrently, UA acts as a damage-associated molecular pattern (DAMP), initiating and sustaining hepatic inflammation. Through dose-dependent activation of the nuclear factor kappa-B (NF-κB) signaling pathway, UA enhances proinflammatory cytokine expression [27] while also promoting macrophage recruitment and NLRP3 inflammasome assembly [28]. This inflammatory cascade leads to robust production of interleukin-1β (IL-1β), fostering a chronic inflammatory state that promotes hepatocyte injury and fibrosis [29,30]. The metabolic consequences of UA exposure extend to systemic insulin sensitivity, as UA impairs insulin signaling by inhibiting the IRS1/Akt pathway [31]. Furthermore, hyperuricemia may adversely affect pancreatic β-cell function via urate crystal deposition in islets, thereby exacerbating insulin resistance [32]. This metabolic disturbance generates a vicious cycle: insulin resistance promotes lipolysis in adipose tissue, leading to increased delivery of free fatty acids to the liver [33], while compensatory hyperinsulinemia stimulates lipogenesis in hepatocytes [34]. Subsequent lipid overload, coupled with persistent oxidative and inflammatory injury, promotes the accumulation of cytotoxic metabolites and progressive liver damage [35]. Elucidating these underlying mechanisms will be crucial for devising more effective strategies for prevention and treatment [36].

This study employed a sufficiently large sample size and accounted for various potential confounding variables, including age, sex, BMI, smoking, alcohol consumption, and diabetes. However, several limitations should be noted. First, the cross-sectional design precludes definitive conclusions regarding causality between UA and NAFLD. Second, certain self-reported confounding factors may be susceptible to recall bias. Third, our study may not have comprehensively accounted for other confounding variables, such as genetic predisposition, dietary intake, environmental exposure, and occupational hazards, which could influence the observed associations.

## 5. Conclusions

In summary, this study identified a significant association between serum UA levels and NAFLD in a hypertensive population. Uric acid may serve as a useful biomarker for predicting NAFLD risk in these patients. Our findings provide a reliable basis for assessing NAFLD risk based on UA levels. Consequently, managing UA levels could have important implications for preventing NAFLD development in hypertension, suggesting that urate-lowering therapy may represent a novel strategy for NAFLD prevention and treatment in patients with hypertension.

## Acknowledgments

We would like to acknowledge Yan F *et al.* for their research and for providing anonymous, publicly available data in the database.

## Author contributions

**Data curation:** Yan Zhao, Ao Gao, Lei Cao.

**Methodology:** Ting Yang, Zhuang Li.

**Project administration:** Yan Zhao, Zhenwei Wang, Yang Liu.

**Writing – original draft:** Yan Zhao.

**Writing – review & editing:** Chongyang Ma.

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
