## [Decision Letter · Decision Letter 0]

11 Dec 2025

Dear Dr. Zhao,

Thank you for submitting your manuscript to PLOS ONE. After careful consideration, we feel that it has merit but does not fully meet PLOS ONE’s publication criteria as it currently stands. Therefore, we invite you to submit a revised version of the manuscript that addresses the points raised during the review process.

We look forward to receiving your revised manuscript.

Kind regards,

Aleksandra Klisic

Academic Editor

PLOS One

Journal Requirements:

National Health Commission of the People's Republic of China

5. In the online submission form, you indicated that the data are available upon reasonable request. Extra data can be accessed via the Dryad data repository at http://datadryad.org/with the doi:10.5061/dryad.7d7wm3809.

Reviewers' comments:

Reviewer's Responses to Questions

**Comments to the Author**

1. Is the manuscript technically sound, and do the data support the conclusions?

Reviewer #1: Yes

Reviewer #2: Yes

2. Has the statistical analysis been performed appropriately and rigorously?

Reviewer #1: Yes

Reviewer #2: Yes

3. Have the authors made all data underlying the findings in their manuscript fully available?

Reviewer #1: Yes

Reviewer #2: Yes

4. Is the manuscript presented in an intelligible fashion and written in standard English?

Reviewer #1: Yes

Reviewer #2: Yes

Reviewer #1: The paper is mostly well composed, organized logically, and simple to understand. The research question is well articulated, and the reasoning for investigating uric acid as a possible risk factor for NAFLD in this group is strongly supported. The results align with current literature and provide further perspective on a particular clinical subgroup that is rarely explored in depth.

this is a thoughtfully structured and meticulously executed research. The findings are reliable, the examination is fitting, and the document is well organized. It satisfies PLOS ONE's criteria for methodological rigor, ethical clarity, and reporting transparency.

I suggest small changes before approval. Particularly:

Modify the data availability statement to align with the requirements of PLOS ONE.

Include a brief comment on model diagnostics and discuss the minimal practical improvement in AUC.

Fix small typographical and formatting errors.

After these minor adjustments, the manuscript will be prepared for publication and will significantly enhance the literature regarding metabolic risk factors among hypertensive groups

Reviewer #2: This is a very interesting study. It was also thoroughly done. The data was well tabulated and also well interpreted. I enjoyed reading every part of this research article. Well done.

Just one thing to point out; the last sentence of the penultimate paragraph in the discussion section doesn't have a correlating reference. It therefore should be rephrased to prevent flagging it for plagiarism and the corresponding reference should be included.

Thank you.

**Do you want your identity to be public for this peer review?** For information about this choice, including consent withdrawal, please see our Privacy Policy

Reviewer #1: **Yes:** Malak M Abdulqadir (Alagoury)

Reviewer #2: **Yes:** Dr. Temiloluwa Adefusi

---

## [Author Response · Author response to Decision Letter 1]

5 Jan 2026

Dear Dr. Klisic and Reviewers,

Thank you for the opportunity to revise our manuscript (PONE-D-25-57606) entitled “Association between uric acid and non-alcoholic fatty liver disease in patients with hypertension” for consideration in PLOS ONE. We sincerely appreciate the editors and reviewers for their time, insightful comments, and constructive suggestions, which have significantly strengthened our manuscript. We have carefully addressed all the points raised, and the corresponding changes have been incorporated into the revised manuscript. A point-by-point response is provided below.

Response to Journal Requirements:

1. Style Requirements:

We have carefully reviewed and formatted our manuscript to fully comply with PLOS ONE’s style requirements, including the use of the provided templates for the main body and title/authors/affiliations sections.

2. & 3. Funding Information and Financial Disclosure:

We apologize for the initial discrepancy. The correct and complete funding statement is as follows:

This research was sponsored by the National Key Research and Development Program of China (No. 2024YFC3506301), and Wang Qi's National Master of Traditional Chinese Medicine Inheritance Studio (DFRCYJ-2024A-001). The funders had no role in study design, data collection and analysis, decision to publish, or preparation of the manuscript.

This statement has been updated in the manuscript file. The authors received no specific funding for this work. And no author received a salary from these funders for conducting this study.

4. Ethics Statement:

The ethics statement has been reviewed and now appears only in the Methods section of the manuscript. Any duplicate statements in other sections have been removed.

5. Data Availability:

The Data Availability Statement in the manuscript has been updated as below to clearly reflect this public deposition, ensuring full compliance with PLOS ONE’s data policy.

The data that support the findings of this study are publicly available from the Dryad repository. These data can be accessed directly, without any special permissions, via the official repository: https://doi.org/10.5061/dryad.7d7wm3809.

6. & 7. References:

We have reviewed the reviewers' comments and confirm that no specific recommendations to cite previously published works were made. We have thoroughly reviewed our reference list, ensuring it is complete, correct, and contains no retracted papers.

Response to Reviewer Comments:

Reviewer #1: We thank the reviewer for the positive assessment and valuable suggestions.

1) Data Availability Statement: We have revised the Data Availability Statement in the manuscript to explicitly state: “The data that support the findings of this study are publicly available from the Dryad repository. These data can be accessed directly, without any special permissions, via the official repository: https://doi.org/10.5061/dryad.7d7wm3809.” This aligns with PLOS ONE’s requirements.

2) Include a brief comment on model diagnostics and discuss the minimal practical improvement in AUC: We sincerely thank the reviewer for this crucial suggestion. We have now added a comprehensive new subsection (3.5 Robustness analysis and model diagnostics) in the Results. This section details:

Model Diagnostics and Robustness:

We have conducted a comprehensive diagnostic evaluation of the final logistic regression model to ensure the robustness and reliability of the results, which includes the following:

(1) Core Parameter Stability (Bootstrap Resampling Validation): To directly quantify the uncertainty of model parameters under sampling variability, we employed the Bootstrap method (1,000 replicates) to calculate robust standard errors and confidence intervals. The results show that the independent positive association between the key variable uric acid (UA) and NAFLD is highly stable (B = 0.002, Bootstrap 95% CI: 0.000 – 0.004, P = 0.019). This interval excludes zero and is precise, serving as the core evidence for the robustness of our conclusion.

(2) Model Goodness-of-Fit: The Hosmer-Lemeshow test indicated that the full model incorporating UA demonstrated good fit (χ² = 11.668, P = 0.167).

Multicollinearity: The variance inflation factor (VIF) for all independent variables was below 10, with the VIF for the key variable UA being 1.248. This indicates the absence of severe multicollinearity, allowing for a clear estimation of the independent effects of UA.

Outliers and Influential Points: A sensitivity analysis was performed after identifying and excluding 55 potential outliers. The direction, magnitude, and statistical significance of the association between UA and NAFLD remained unchanged (OR = 1.003, 95% CI: 1.001–1.005, P = 0.014). This confirms that our primary finding is not driven by a small number of extreme data points and possesses generalizability.

(3) Conclusion: The aforementioned analyses form a complete chain of evidence. Bootstrap validation confirms the stability of the statistical inference, the H-L test and collinearity diagnostics ensure the rationality of the model construction, and the sensitivity analysis rules out the excessive influence of outliers. Together, they collectively support the high robustness of the study's conclusions.

In the Discussion, we have integrated a dedicated comment (in the first paragraph) addressing the minimal practical improvement in AUC. We acknowledge the modest absolute increase (from 0.770 to 0.772) and discuss its practical value in the context of UA being a routine, low-cost test in hypertension management, thus offering a cost-effective enhancement to risk stratification.

3) Typographical and Formatting Errors: We have meticulously proofread the entire manuscript and corrected the typographical and formatting errors.

Reviewer #2: We thank the reviewer for their encouraging feedback and for identifying the oversight.

Reference in the Discussion: We have rephrased the final sentence of the penultimate paragraph in the Discussion section and have included the appropriate citation ([36]) to support the statement regarding future research directions for elucidating pathophysiological mechanisms. This corrects the previous omission and ensures proper attribution.

Conclusion

We believe that all concerns raised have been adequately addressed. The revisions have substantially strengthened the clarity and coherence of the presentation, as well as the overall rigor of the scientific argument. We have uploaded the following files via the submission system:

1. Response to Reviewers (this letter).

2. Revised Manuscript with Track Changes (highlighting all modifications).

3. Manuscript (clean version without tracked changes).

Thank you again for your consideration. We look forward to your decision on our revised manuscript.

Sincerely,

---

## [Decision Letter · Decision Letter 1]

15 Jan 2026

Association between uric acid and non-alcoholic fatty liver disease in patients with hypertension

PONE-D-25-57606R1

Dear Dr. Zhao,

We’re pleased to inform you that your manuscript has been judged scientifically suitable for publication and will be formally accepted for publication once it meets all outstanding technical requirements.

Kind regards,

Aleksandra Klisic

Academic Editor

PLOS One

Additional Editor Comments (optional):

Reviewers' comments:

Reviewer's Responses to Questions

**Comments to the Author**

Reviewer #2: All comments have been addressed

2. Is the manuscript technically sound, and do the data support the conclusions?

Reviewer #2: Yes

3. Has the statistical analysis been performed appropriately and rigorously?

Reviewer #2: Yes

4. Have the authors made all data underlying the findings in their manuscript fully available?

Reviewer #2: Yes

5. Is the manuscript presented in an intelligible fashion and written in standard English?

Reviewer #2: Yes

Reviewer #2: All comments have been addressed, the article is more concise and easier to read and plagiarisms have been sorted. Well done

**Do you want your identity to be public for this peer review?** For information about this choice, including consent withdrawal, please see our Privacy Policy

Reviewer #2: **Yes:** ADEFUSI TEMILOLUWA

---

## [Editor Report · Acceptance letter]

PONE-D-25-57606R1

PLOS One

Dear Dr. Zhao,

I'm pleased to inform you that your manuscript has been deemed suitable for publication in PLOS One. Congratulations! Your manuscript is now being handed over to our production team.

Kind regards,

on behalf of

Dr. Aleksandra Klisic

Academic Editor

PLOS One